# PARP10 Multi-Site Auto- and Histone MARylation Visualized by Acid-Urea Gel Electrophoresis

**DOI:** 10.3390/cells10030654

**Published:** 2021-03-15

**Authors:** Antonio Ginés García-Saura, Herwig Schüler

**Affiliations:** 1Department of Biosciences and Nutrition, Karolinska Institutet, 14157 Huddinge, Sweden; antonio.gines.garcia.saura@ki.se; 2Center for Molecular Protein Science, Department of Chemistry, Lund University, 22100 Lund, Sweden

**Keywords:** acid-urea PAGE, ADP-ribosylation, ARTD10, histone H3, MARylation, mass spectrometry, PARP10

## Abstract

Poly-ADP-ribose polymerase (PARP)-family ADP-ribosyltransferases function in various signaling pathways, predominantly in the nucleus and cytosol. Although PARP inhibitors are in clinical practice for cancer therapy, the enzymatic activities of individual PARP family members are yet insufficiently understood. We studied PARP10, a mono-ADP-ribosyltransferase and potential drug target. Using acid-urea gel electrophoresis, we found that the isolated catalytic domain of PARP10 auto-ADP-ribosylates (MARylates) at eight or more acceptor residues. We isolated individual species with either singular or several modifications and then analyzed them by mass spectrometry. The results confirmed multi-site MARylation in a random order and identified four acceptor residues. The mutagenesis of singular acceptor residues had a minor impact on the overall auto-MARylation level and no effect on the MARylation of histone H3.1. Together, our results suggest that PARP10 automodification may have functions in the regulation of intramolecular or partner binding events, rather than of its enzymatic catalysis. This contributes to a better understanding of PARP10 functions, and, in the long run, to gauging the consequences of PARP inhibitor actions.

## 1. Introduction

Single genes encode multiple differentially functional proteins. While the human genome contains an estimated 25,000 genes, the human proteome consists of more than one million different polypeptides [1]. This increase in complexity is possible thanks to, among other mechanisms, protein posttranslational modifications (PTMs), covalent alterations that allow for the swift modulation of enzymatic activities and subcellular localizations. Similar to other PTMs, ADP-ribosylation has gained attention for its vital importance in cell homeostasis: The discovery of the role of PARP1 dependent poly-ADP-ribosylation (PARylation) in DNA single-strand break repair [2,3], and the following application of PARP inhibitors in the treatment of BRCA1/2 deficient cancers [4], have catalyzed a growing interest in PARylation research.

Despite the family name, most members of the poly-ADP-ribose polymerase (PARP) family are mono-ADP-ribosyltransferases (mARTs). These enzymes catalyze the transfer of a single unit of ADP-ribose to their target, consuming a molecule of nicotinamide adenine dinucleotide (NAD^+^) in the process. They can transfer ADP-ribose to a wide array of targets, including glutamate, aspartate, serine, lysine, and arginine side chains in peptides and proteins [5,6], and to phosphorylated ends of RNA [7]. The arguably most well-studied enzymes in this sub-family are PARP10/ARTD10 and PARP14/ARTD8/BAL2; but we are only beginning to understand their substrate specificities and biological implementations.

Recent proteomics studies have made an important contribution to understanding the scale of protein MARylation. The in vitro PARP10 dependent MARylome in HEK293T cells may comprise over 800 specific target proteins [8], many of which have been identified also in other systems and by other methods [9,10,11]. These studies suggest that PARP10 is involved in a wide array of pathways and functions. Two groups of proteins are especially abundant among the PARP10 targets; those that are involved in ubiquitin transfer and those linked to mRNA regulation. These findings were not unexpected, given that PARP10 contains ubiquitin interaction [12] and RNA recognition motifs [13], and they are supported by previous results [12]. Nevertheless, the vast majority of putative PARP10 targets have not been verified and the consequences of their modification remain largely undefined. PARP10 has automodification activity [14,15], like other PARP family members; but it stands out with its high levels of automodification in vitro [16]. Several sites of automodification have been identified [14,17,18,19]. However, the significance of auto-MARylation is currently not understood.

The verification of poly-ADP-ribosylation (PARylation) mediated by PARP1 and similar enzymes is straightforward *in vitro*, as their activities produce a prominent shift in electrophoretic mobility, which is visible as a “smear” in SDS-PAGE gels stained with Coomassie [15]. In MARylation, by contrast, the resulting mass difference is too small for this method to be useful. Consequently, the in-gel detection of MARylation requires the use of modified substrate—either ^32^P-NAD^+^, making laboratory work and analysis more demanding; or biotinylated and fluorescent analogs of NAD^+^, introducing the risk of artefacts due to the chemical modifications. However, acidic modifications of proteins can be visualized by exploiting the altered charge in electrophoretic mobility assays. For instance, reverse-polarity PAGE using TBE gels has been used to separate MARylated and otherwise modified histone tail peptides [20]. Furthermore, in acid-urea polyacrylamide gel electrophoresis (AU-PAGE), histone proteins that carry multiple modifications are resolved into distinct bands to produce a “histone ladder” [21].

In this study, we developed an AU-PAGE protocol to visualize MARylation of basic proteins, in which the addition of two negative charges per ADP-ribosyl unit reduces the mobility of the urea denatured proteins as they migrate toward the cathode. Our AU-PAGE protocol allowed for the visualization of trans-MARylation of histone proteins as well as auto-MARylation at several sites of a basic PARP10 catalytic domain construct. We verified auto-MARylation by mass spectrometry. The mutagenesis of individual acceptor residues had no apparent impact on the modification of others, leading us to suggest that PARP10 auto-MARylation is not directly involved in the regulation of PARP10 catalytic activity.

## 2. Materials and Methods

### 2.1. Reagents and Proteins

Chemicals were purchased from Sigma–Aldrich, unless otherwise noted. Electrophoresis reagents were from ThermoFisher Scientific, Gothenburg, Sweden. Recombinant human histone H3.1 was purchased from New England Biolabs (M2503). Expression vectors for the PARP10 catalytic domain constructs N819-V1007 (wild type and mutants) were constructed by sub-cloning synthetic DNA fragments (GeneArt; ThermoFisher Scientific) into pNIC-28 [22]. The cDNAs encoding human MacroD2^21–245^ and human TARG1/C6orf130^1–152^ were inserted into pNIC-28 to obtain N-terminal hexahistidine fusions. Human ARH3 expression plasmid (full length sequence in pET26) [23] was kindly contributed by Friedrich Koch-Nolte (University Medical Center Hamburg-Eppendorf, Hamburg, Germany). All of the proteins were expressed in *Escherichia coli* strain BL21(DE3)T1R (Sigma–Aldrich, Stockholm, Sweden) and then purified by immobilized metal affinity chromatography, followed by size exclusion chromatography.

### 2.2. Enzymatic Reactions

For the automodification reaction, 10 μM of the PARP10 catalytic domain constructs were incubated with the corresponding amount of NAD^+^ in reaction buffer (50 mM HEPES, 100 mM NaCl, 0.2 mM TCEP, 4 mM MgCl_2_, and pH 7.5) for 30 min. NAD^+^ analogs 6-fluoresceine-10-NAD^+^ (6-F-NAD^+^) [24] and 8-butylthio-NAD^+^ (8-but-NAD^+^) [25] were purchased from Biolog (Bremen, Germany); nicotinamide-N6-ethenoadenosine dinucleotide (εNAD^+^) [26] was purchased from Sigma–Aldrich. For trans-MARylation, 10 µM of the catalytic domain of PARP10 or the different mutants were co-incubated with 10 µM of histone H3.1 (M2503S; New England BioLabs/BioNordika, Stockholm, Sweden) in the presence of 1 mM NAD^+^ in reaction buffer for 30 min. For ADP-ribosyl glycohydrolase reactions, 10 µM of the corresponding enzyme (human ARH3, MacroD2 and TARG1) were added to automodified PARP10 catalytic domain in reaction buffer and then incubated for 60 min. under constant shaking. All of the reactions were carried out at ambient temperature (22 °C).

### 2.3. General Electrophoresis

SDS-PAGE was carried out using NuPAGE 4–12% Bis-Tris gels and MES containing running buffer (ThermoFisher Scientific). TBE-urea PAGE was done using Novex 6% TBE-urea gels, as recommended by the manufacturer (ThermoFisher Scientific). All types of gels were stained with Coomassie using the method described by Yasumitsu [27].

### 2.4. Acid-Urea PAGE

Appendix B provides a detailed proceeding of AU-PAGE. Briefly, a resolving gel containing 4 M urea, 15% acrylamide, and 5% acetic acid, and a stacking gel containing 8 M urea, 7.5% acrylamide, and 5% acetic acid were cast in a SureCast system (ThermoFisher Scientific). The gel was pre-run overnight before the samples were separated in 5% acetic acid at a constant current of 15 mA. The gel bands in Figure 3 were analyzed by estimating total intensities in equal areas for each condition using ImageJ software (vs. 1.53e, NIH, Bethesda, MD, USA) [28].

### 2.5. Mass Spectrometry

In-gel Protein Digestion—protein bands were excised from Coomassie-stained gels and then digested in-gel using a MassPREP robotic protein-handling system (Waters, Millford, MA, USA). The gel slices were destained twice with 100 µL of 50 mM ammonium bicarbonate (Ambic) containing 50% acetonitrile (ACN) at 40 °C for 10 min. Proteins were then reduced with 10 mM dithiothreitol (DTT) in 100 mM Ambic for 30 min. at 40 °C and then alkylated with 55 mM iodoacetamide in 100 mM Ambic for 20 min. at 40 °C, followed by digestion with 0.3 µg trypsin (sequencing grade, Promega, Madison, WI, USA) in 50 mM Ambic for 5 h at 37 °C. The tryptic peptides were extracted with 1% formic acid (FA) in 2% ACN, followed by 50% ACN twice. The solutions were dried, and the peptides were cleaned up on C-18 Stage Tips (ThermoFisher Scientific).

Liquid Chromatography-Tandem Mass Spectrometry—the reconstituted peptides in solvent A (0.1% FA in 2% ACN) were separated on a 50 cm EASY-Spray C18 column (ThermoFisher Scientific) that was connected to an Ultimate-3000 nano-LC system (ThermoFisher Scientific) using a 60 min. gradient from 2–26% of solvent B (98% AcN, 0.1% FA) in 55 min. and up to 95% of solvent B in 5 min. at a flow rate of 300 nL/min. Mass spectra were acquired on an Orbitrap Fusion Lumos tribrid mass spectrometer (ThermoFisher Scientific) in *m/z* 300 to 1750 at resolution of R = 120,000 (at *m*/*z* 200) for full mass, followed by data-dependent higher energy collisional dissociation (HCD) fragmentations of precursor ions with a charge state 2+ to 7+ in 3 s cycle time [11]. The tandem mass spectra were acquired with a resolution of R = 30,000, targeting 5 × 10^5^ ions.

Data analysis—the aquired raw data files were analyzed using Proteome Discoverer v2.4 (ThermoFisher Scientific) with Sequest HT and MS Amanda 2.0 search engines against a customized protein database that included Q53GL7 (SwissProt, Lausanne, Switzerland) among other 45 sequences. A maximum of two missed cleavage sites were allowed for trypsin, while setting the precursor and the fragment ion mass tolerance to 10 ppm and 0.02, respectively. The carbamidomethylation of cysteine was specified as a fixed modification. ADP-ribosylation of arginine, asparagine, aspartic acid, glutamic acid, glutamine, lysine, serine, threonine and cysteine, methionine oxidation, and acetylation of N-termini were set as dynamic modifications. The initial search results were filtered with 5% FDR using the fix value PSM validator node and IMP-ptmRS [29] was used to localize and calculate the scores of variable modifications.

## 3. Results and Discussion

### 3.1. PARP10 Auto-MARylation Can Be Visualized by Acid-Urea Polyacrylamide Gel Electrophoresis (AU-PAGE)

We set out to develop a qualitative PAGE assay for PARP10 catalytic activity. Although the auto-MARylation of the PARP10 catalytic domain results in an increase of its apparent molecular weight (Figure 1a), SDS-PAGE lacks the resolution that is required to visualize a mobility shift. However, the constructs of the PARP10 catalytic domain are basic proteins. Therefore, we reasoned that the addition of negative charges due to auto-MARylation should be resolved by inherent charge separation. Indeed, auto-MARylated PARP10 produced a prominent smear in TBE-urea gels, whereas the unmodified protein stayed on top of the gel (Figure 1a). Furthermore, when PARP10 auto-MARylation reaction mixtures were separated by acid-urea gel electrophoresis (Appendix B), unmodified and modified proteins were resolved (Figure 1). When PARP10 was treated with 1 mM NAD^+^ for 1 h at ambient temperature prior to electrophoresis, AU-PAGE gels showed multiple bands that appeared to indicate a different degree of MARylation. Indeed, treatment with 6-F-NAD^+^ resulted in fewer bands that were more interspaced, apparently indicating less efficient MARylation and a greater mass addition per modification (1178 Da vs. 559 Da).

We turned to a PARP10 target protein to verify that the observed AU-PAGE band pattern reflected multi-site MARylation. Core histones have been shown to be MARylated at several residues [5,19]. Indeed, PARP10 modification of histone H3.1 resulted in a pattern of bands similar to that on PARP10 itself (Figure 1b). No change in mobility was observed for H3.1 that was incubated with 1 mM NAD^+^ in the absence of PARP10. We used the catalytically inactive mutant, PARP10-T912A, to verify that the mobility shift of PARP10 itself was not a consequence of non-enzymatic adducts (Appendix A). The T912A mutant that was incubated with 1 mM NAD^+^ for 1 h at ambient temperature was indistinguishable by AU-PAGE from the control reaction incubated in NAD^+^-free buffer (Figure 1c), which indicated that no non-enzymatic adducts were formed under these conditions.

In order to further confirm these observations, we demonstrated that the formation of PARP10 auto-MARylation band patterns was dependent on the concentration of NAD^+^ supplied (Figure 1d). Efficient multi-site MARylation required NAD^+^ concentrations of 500 µM–1 mM, which is roughly 5–10-fold higher than the *K*_m_^(NAD+)^ for PARP10 ([16] and Appendix A). Finally, we analyzed PARP10 by AU-PAGE after incubation with NAD^+^ and three NAD^+^ analogs (Figure 1e,f). This experiment confirmed that 6-F-NAD^+^, while being a less efficient substrate for PARP10, does not block PARP10 activity: at 1 mM 6-F-NAD^+^, the modification at three sites at most was recognizable. At an equimolar ratio (1 mM in total) of NAD^+^ over 6-F-NAD^+^, the band pattern produced indicated multi-site modification and residual fluorescence from 6-F-NAD^+^ incorporation could be observed. On the contrary, both ε-NAD^+^ and 8-but-NAD^+^ were unable to serve as substrates, and their presence at equimolar ratio prevented the processing of NAD^+^ and auto-MARylation. Thus, we conclude that PARP10 auto-MARylates at least eight distinct positions within its catalytic domain (Figure 1a).

### 3.2. ADP-Ribosyl Glycohydrolases Reduce the Band Shifts Observed with AU-PAGE

PARP10 auto-MARylation is susceptible to hydrolysis by several ADP-ribose erasers [14,30]. We tested the effect of three ADP-ribosyl hydrolases on previously auto-MARylated PARP10: human TARG1 and MacroD2, which both remove MARylation on carboxylic sidechains [30,31,32]; and ARH3, which hydrolyzes ADP-ribosylation on serine [33,34]. All three enzymes were able to partially reduce MARylation of PARP10 (Figure 2). These results indicated that PARP10 auto-MARylation does not occur in a residue specific manner: neither of the three glycohydrolases were able to efficiently remove ADP-ribosyl from PARP10. The latter is consistent with previous observations [17,30,31,32,34,35]. Incidentally, none of these glycohydrolases caused a visible reduction of MARylation when 6-Fluo-NAD^+^ was used as a substrate (Appendix A).

### 3.3. Mass Spectrometry of PARP10 Isolated from AU-PAGE Gels Identifies New Target Residues

Several of the PARP10 auto-MARylation target residues are located in the catalytic domain [14,16,17,18,19,32]. However, it is unknown whether the modification of these residues has functional consequences. To address this question, we first asked whether the modifications of residues appear in a particular order. We auto-MARylated PARP10^819–1007^, subjected it to AU-PAGE, excised gel bands corresponding to protein with different mobility, and processed them for the identification of tryptic peptides by mass spectrometry.

Table 1 summarizes the results of these experiments. We identified four residues as targets for auto-MARylation, namely E825, R855, S857, and E882 (Appendix A). The identification of E825 [17] and E882 [14,17,19] confirms previous observations. However, although the respective peptides were covered in all samples, we were unable to confirm other previously identified residues as targets [14,17,18,19,32]. Because we expected to identify up to 8–10 sites, we assume that experimental and methodological differences are reasons for agreement with previous studies in only two positions.

All four auto-MARylation target residues were identified in several NAD^+^ treated samples, originating from both higher and lower mobility protein. Nevertheless, we believe that E825 may be one of the foremost target residues for PARP10 auto-MARylation: (i) MAR-E825 containing peptides were more abundant than peptides containing modification at any of the other three sites (Table 1). (ii) Perhaps more importantly, E825 was the only target residue that we identified in samples that were treated with the less reactive 6-F-NAD^+^ (Table 1). On the other hand E882, which is the target site that previous studies have identified [14,19], was also identified in several NAD^+^ treated samples.

### 3.4. Three Auto-MARylation Target Residues Are not Essential for Catalytic Activity

Because E825 appeared to emerge as a prominent target residue for PARP10 auto-MARylation, it was interesting to use AU-PAGE to examine the function of this residue further. We generated the mutants E825A, S857A, and E882A of PARP10^819–1007^ and then tested their auto-MARylation products by AU-PAGE (Figure 3). All three mutations resulted in a mild reduction in the number of up-shifted protein bands (Figure 3a). This reduction was the most prominent for E882A, in which case some unmodified PARP10 clearly remained after incubation with NAD^+^ for 1 h. This is also similar to a previous observation using the same mutation [14]. We reasoned that a reduction in mobility of the E882A protein might be caused by either the loss of a prominent acceptor residue, or by kinetic consequences owing to the loss of this residue. To discriminate between these two possibilities, we analyzed trans-MARylation of histone H3.1 (Figure 3b). The result showed that all four proteins were equally able to induce a band pattern in H3.1. We conclude that E882 is a prominent target residue for auto-MARylation; but under our in vitro conditions, auto-MARylation of neither E825, S857, nor E882 has a direct regulatory function in the catalytic activity of PARP10.

## 4. Conclusions

In the present work, we applied AU-PAGE to the study of protein ADP-ribosylation. The resolution of auto-MARylated PARP10 catalytic domain allowed for the identification of species of increasing molecular weight and negative charge. Mass spectrometry analysis of these species identified four target residues for auto-MARylation: E825 and E882, as in previous studies; and, R855 and S857, which were identified here for the first time. Although the modification of these residues did not affect catalysis in the catalytic domain construct, a future research goal will be to understand whether auto-MARylation at these sites might modulate the interactions between PARP10 domains, targets, or binding partners.

## Figures and Tables

**Figure 1 cells-10-00654-f001:**
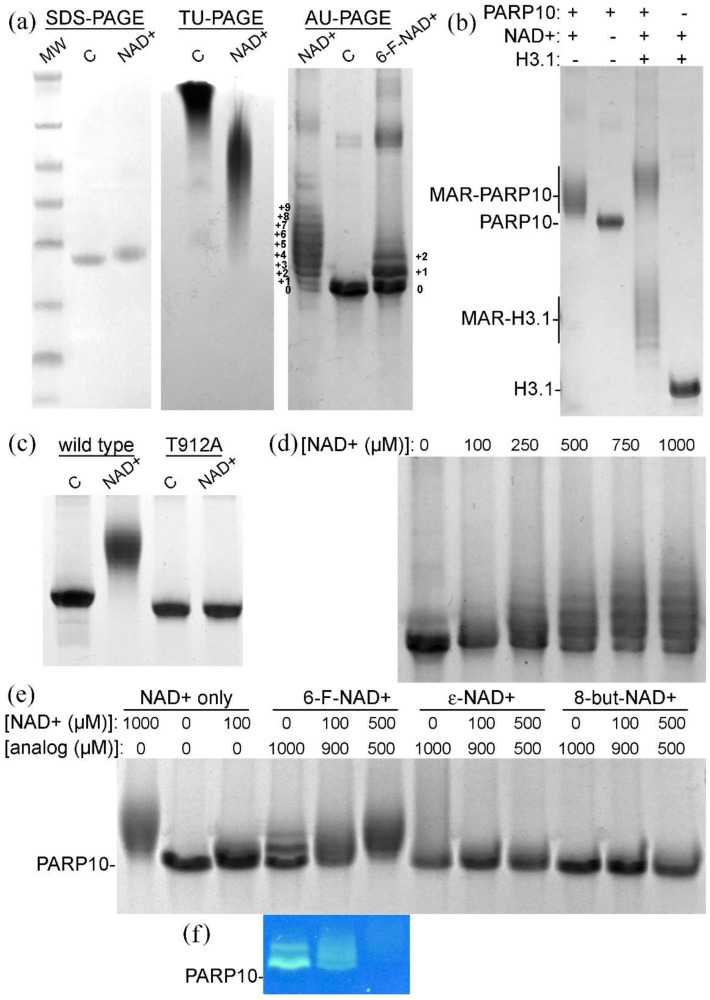
Visualization of PARP10^819–1007^ activity by polyacrylamide gel electrophoresis. (**a**) Coomassie stained SDS-PAGE, TBE-urea PAGE (TU-PAGE), and acid-urea PAGE (AU-PAGE) gels of untreated (C) and auto-MARylated PARP10. MW, molecular weight marker; (**b**) AU-PAGE gel showing PARP10 and histone H3.1 MARylation; (**c**) AU-PAGE gel showing wild type PARP10 and inactive mutant T912A, either untreated or incubated with 1 mM nicotinamide adenine dinucleotide (NAD^+^); (**d**) NAD^+^ concentration dependent mobility shift of PARP10; (**e**) PARP10 auto-MARylated using NAD^+^ alone or in combination with the indicated NAD^+^ analogues; (**f**) fluorescence imaging of the gel area containing 6-F-NAD^+^ treated protein; and, (**e**,**f**) the positions of unmodified PARP10 are marked.

**Figure 2 cells-10-00654-f002:**
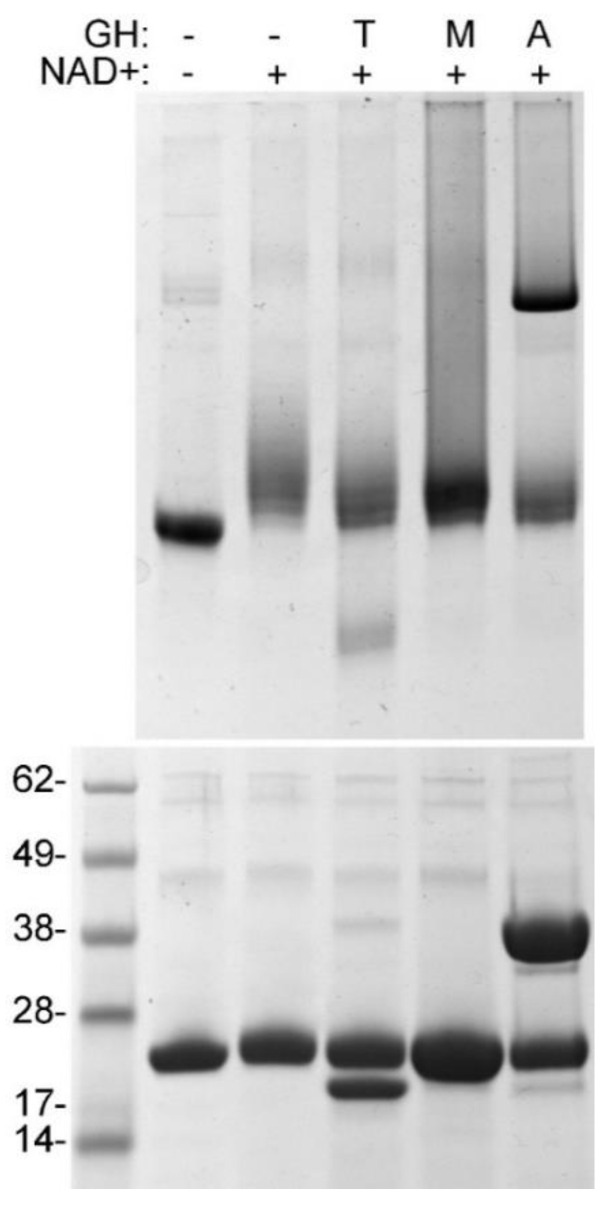
AU-PAGE analysis of ADP-ribosyl glycohydrolase action on auto-MARylated PARP10. PARP10^819–1007^ (10 µM) was incubated sequentially with NAD^+^ and with either TARG1 (T), MacroD2 (M), or ARH3 (A) at equimolar concentration as described to detail under Materials and Methods. Reactions were divided and processed by AU-PAGE (**upper panel**) and SDS-PAGE (**lower panel**).

**Figure 3 cells-10-00654-f003:**
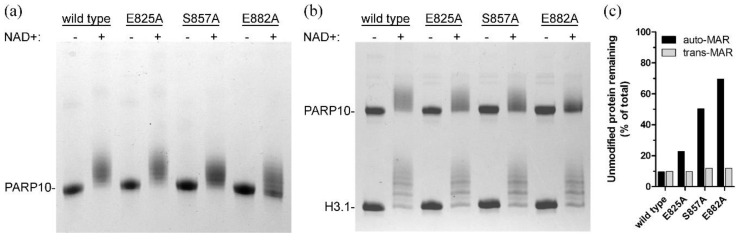
AU-PAGE analysis of wild type and mutant PARP10^819–1007^ activities. (**a**) Auto-MARylation; (**b**) Histone H3.1 MARylation. (**c**) Estimates of the percentages of remaining unmodified protein in the NAD^+^ containing gel lanes (black columns, unmodified PARP10; grey columns, unmodified H3.1).

**Table 1 cells-10-00654-t001:** Identification of PARP10 auto-MARylation sites by mass spectrometry ^1^.

Res.	Peptide Sequence	PARP10 + NAD^+ 2^			
Gel Band		Upper	Mid	Lower	All
E825	823-LA**E**NTGEFQEVVR-835	98% (×5)	96% (×6)	97% (×4)	99% (×7)
R855	853-VE**R**VSHPLLQQQYELYR-869	100% (×1)	100% (×1)	100% (×1)	100% (×1)
S857	853-VERV**S**HPLLQQQYELYR-869	99% (×1)	98% (×1)	-	100% (×1)
E882	879-RPVEQVLYHGTTAPAVPDICAHGFNR-904	100% (×1)	100% (×1)	-	100% (×1)

^1^ Bands were excised from AU-PAGE gels and processed for peptide mass identification. Where indicated, the upper (6 and more modifications), mid (3–5 modifications) and lower (0–2 modifications) bands were excised and analyzed separately. Shown are the probabilities of residue identification, and the number of occurrences in parentheses. ^2^ Only ADPr-E825 was identified when 6-F-NAD^+^ was used as substrate (upper band: 98%, ×1; all bands: 100%, ×1). See Appendix A for LC-MS/MS spectra.

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
