# Peer review of "PARP10 Multi-Site Auto- and Histone MARylation Visualized by Acid-Urea Gel Electrophoresis"

_cells, 2021, doi:10.3390/cells10030654_

Round 1

Reviewer 1 Report

Saura and Schuler proposed the use of Acid-Urea Gel electrophoresis in analyzing MARylation and identified the site by Mass spectrometry. The manuscript would be of interest to the field and should be considered for publication with the following improvements:

(1) The technique can only be performed on basic protein— a minor caveat of the technique, which should be emphasized. Can the authors comment on how their technique compares with the one developed by Ivan Matic's group (PMID: 33186521 and 30267210)?

(2) Can the authors comment on why Cysteine was not searched? Also, how did they assign the ADP-ribose site? Could they also show the mass spectra?

(3) Could the author quantitate the data on Figure 3a and b (e.g. using a line profiler) to indicate the differential effect of mutants on automodification vs. trans-modification? This comparison is one of the highlights of the manuscript, which will be reinforced by the quantitation to demonstrate their point that the automodification does not affect the PARP 10 activity

(4) The citation accuracy needs to be improved:

(a) Line 43-46: the comparison between PARP1/2 vs. PARP10 is not fair as they are not performed in parallel or the same method.

(b) PARP10 substrates were also defined by proteome array (PMID: 23332125)

(c) PARP10 sites were also identified in two additional studies (PMID: 27406238 and 28143925 [E825 also identified in this study]) 

(5) Line 182-183: Can the authors show the data?

Author Response

Saura and Schuler proposed the use of Acid-Urea Gel electrophoresis in analyzing MARylation and identified the site by Mass spectrometry. The manuscript would be of interest to the field and should be considered for publication with the following improvements:

We thank this reviewer for their time and effort.

(1) The technique can only be performed on basic protein— a minor caveat of the technique, which should be emphasized.

It is true that the separation required to visualize individual ADPr is best achieved with proteins with a basic pI in physiological conditions. Nevertheless, under the conditions of electrophoresis (i.e. pH 2.3) a majority of proteins will migrate into the gel. For example: Figure 2 - several of the glycohydrolases (around neutral pI at physiological pH) resolve well. Naturally, electrophoretic mobility is dependent on several factors including charge-to-size ratio and resistance to urea induced unfolding. Therefore, we believe that a statement according to the reviewer’s suggestion would make readers believe that AU-PAGE is not suited for proteins with a neutral or slightly acidic pI, which is not true. Taken together, we trust that readers will understand that separation of proteins other than PARP10 and histones will require some method adjustment and we do not believe that further clarification is needed.

Can the authors comment on how their technique compares with the one developed by Ivan Matic's group (PMID: 33186521 and 30267210)?

The PAGE method developed by the Matic group employs TBE gels run under non-denaturing conditions with reverse polarity to separate ADP-ribosylated from unmodified (or otherwise modified) histone tail peptides. To the best of our knowledge, the method has not been used for proteins.

To test the reviewer’s suggestion, we used commercial TBE-urea gels, sample and running buffers. Unmodified and auto-MARylated PARP10 showed distinct migration patterns (our new panel in Figure 1a) but we have been unable to identify conditions under which the “smear” resolves into single bands. We have also added reference to the Bartlett et al paper (PMID: 30267210) and expanded the text with a brief description of TBE-urea PAGE.

(2) Can the authors comment on why Cysteine was not searched?

We thank the reviewer for spotting this mistake, which we have corrected in the revised methods description. All cysteine containing peptide masses were consistent with methyl-cysteines (due to the use of alkylating agent in the protocol).

Also, how did they assign the ADP-ribose site? Could they also show the mass spectra?

We now show examples of mass spectra for peptides containing each of the target residues as Supplementary Figure S3.

(3) Could the author quantitate the data on Figure 3a and b (e.g. using a line profiler) to indicate the differential effect of mutants on automodification vs. trans-modification? This comparison is one of the highlights of the manuscript, which will be reinforced by the quantitation to demonstrate their point that the automodification does not affect the PARP 10 activity

We used ImageJ software to estimate the amounts of unmodified proteins remaining in the presence of NAD+ for wild type PARP10 and the three mutant proteins, in either auto- or trans-modification. The results (new figure panel: 3c) confirm the visual impression and make the effect of the mutation easier to grasp for the reader. We thank the reviewer for this suggestion.

(4) The citation accuracy needs to be improved:

(a) Line 43-46: the comparison between PARP1/2 vs. PARP10 is not fair as they are not performed in parallel or the same method.

Although the same general method was used by the same (Cohen’s) laboratory, we agree that a numerical comparison of the results of the different studies is misleading. We have deleted this discussion, which does not change the general message, namely, that PARP10 appears to have a wide range of targets.

(b) PARP10 substrates were also defined by proteome array (PMID: 23332125)

Although we specifically focused this discussion on the recent proteome analyses, we agree that this would be a proper place to cite similar studies and have added a reference to the paper by Feijs et al, 2013, as well as another recent study profiling PARP10 targets (Saei et al., 2021).

(c) PARP10 sites were also identified in two additional studies (PMID: 27406238 and 28143925 [E825 also identified in this study]) 

We thank the reviewer for refreshing our memories about the contents of these studies. We have included reference to them in the introduction as well as in section 3.3, and made minor text changes regarding previously known target residues.

(5) Line 182-183: Can the authors show the data?

As we believe that this will be of interest to a limited readership, we have integrated a figure in our new Supplementary Materials as Figure S2. The gel shows the same reactions as Figure 2 but conducted in the presence of 1 mM 6-F-NAD+; and we show a Coomassie stain and a fluorescence image of the AU-PAGE gel.

Reviewer 2 Report

This is an elegant work providing important details about auto- and trans-MARylation of PARP10 catalytic domain.

The presented work is well done with thorough controls and fairly straightforward interpretations of the data. Nevertheless, I have some minor comments that might increase the strength of the manuscript:

  1. Section 3.1. In order to exclude the potential contribution of non-catalytic adducts in ADP-ribosylation sites identified in this study, the authors should consider employing the catalytic-inactive mutant of PARP10 incubated with NAD+ as an additional control.
  2. Section 3.2 and/or Section 4. A more detailed speculation about the inefficient hydrolysis of ADP-ribosylated PARP10 by TARG1, MacroD2 and ARH3 glycohydrolases should be provided.
  3. Table 1. I would like to suggest introducing an additional column to the present table that illustrates the extended amino acid sequence of ADP-ribosylated peptides. Authors may also comment in the text on potential modification consensus.

Author Response

This is an elegant work providing important details about auto- and trans-MARylation of PARP10 catalytic domain.

We thank also this reviewer for their time, effort, and praise.

The presented work is well done with thorough controls and fairly straightforward interpretations of the data. Nevertheless, I have some minor comments that might increase the strength of the manuscript:

  1. Section 3.1. In order to exclude the potential contribution of non-catalytic adducts in ADP-ribosylation sites identified in this study, the authors should consider employing the catalytic-inactive mutant of PARP10 incubated with NAD+ as an additional control.

The experiment displayed in Figure 1b shows that no non-enzymatic adducts appear on H3.1 when incubated with 1 mM NAD+ as a control. Nevertheless, we thank you for this suggestion as the proposed experiment will strengthen the quality of our study.

The mutant G888W of PARP10 has been widely used as a control for catalytic activity. Unfortunately, our structural work suggests that the introduced tryptophan sidechain disturbs the hydrophobic core of the domain and we have independent biochemical and biophysical evidence that the PARP10-G888W catalytic domain is partially unfolded. Therefore we have chosen in our laboratory to not rely upon the G888W mutant.

Another residue that has been mutated to compromise PARP10 catalytic activity is I987. We analyzed the mutants I987G and I987D. Both have residual activity (with a KM in the low millimolar range, according to our unpublished analyses) and produce a smear (but no “ladder”) in AU-PAGE. Thus, also this mutant was not suitable to test the occurrence of non-enzymatic adducts on PARP10.

To identify an alternative mutation for control reactions, we have previously mutated every residue in two active-site loops, the A- and D-loops, and characterized the stability and kinetic properties of the mutant proteins. While the results are still unpublished, we present proof here for the catalytic inactivity of the mutant T912A of PARP10 (KM data of wild type and mutant in new Supplementary Figure S1) and we have added a new panel Figure 1c showing an AU-PAGE gel of wild type and mutant PARP10 processed under our regular conditions, showing no modification of the mutant protein.

  1. Section 3.2 and/or Section 4. A more detailed speculation about the inefficient hydrolysis of ADP-ribosylated PARP10 by TARG1, MacroD2 and ARH3 glycohydrolases should be provided.

All three glycohydrolases have previously been shown to only partially remove ADPr from its targets. This is likely due to their side chain specificity. We have expanded section 3.2 with a statement and reference to six previous studies that our observations are consistent with.

  1. Table 1. I would like to suggest introducing an additional column to the present table that illustrates the extended amino acid sequence of ADP-ribosylated peptides. Authors may also comment in the text on potential modification consensus.

We thank the reviewer for this suggestion. We have added the peptide sequences in the table, and simplified the table by removing the information about 6-F-ADP-ribosyl peptides into the figure legend. We believe that these changes improve clarity.

The question of a potential modification consensus is without doubt highly interesting. As far as we know, a pattern for target residue recognition has never been recognized for any PARP enzyme. We feel that a discussion in lieu of experimental validation would not add value to our manuscript.

Round 2

Reviewer 2 Report

The authors addressed all my concerns. I am happy with the revised and greatly improved version of the manuscript, which is now ready for publication in Cells.